# The Interactions of H_2_TMPyP, Analogues and Its Metal Complexes with DNA G-Quadruplexes—An Overview

**DOI:** 10.3390/biom11101404

**Published:** 2021-09-25

**Authors:** Catarina I. V. Ramos, Ana R. Monteiro, Nuno M. M. Moura, Maria Amparo F. Faustino, Tito Trindade, Maria Graça P. M. S. Neves

**Affiliations:** 1LAQV-REQUIMTE, Department of Chemistry, University of Aveiro, 3810-193 Aveiro, Portugal; anarita.rvcm@ua.pt (A.R.M.); nmoura@ua.pt (N.M.M.M.); faustino@ua.pt (M.A.F.F.); gneves@ua.pt (M.G.P.M.S.N.); 2CICECO-Aveiro, Institute of Materials, Department of Chemistry, University of Aveiro, 3810-193 Aveiro, Portugal; tito@ua.pt

**Keywords:** aromatic ligands, porphyrins, metalloporphyrins, H_2_TMPyP, Ag^II^TMPyP, G-quadruplexes, telomerase inhibition, selectivity

## Abstract

The evidence that telomerase is overexpressed in almost 90% of human cancers justifies the proposal of this enzyme as a potential target for anticancer drug design. The inhibition of telomerase by quadruplex stabilizing ligands is being considered a useful approach in anticancer drug design proposals. Several aromatic ligands, including porphyrins, were exploited for telomerase inhibition by adduct formation with G-Quadruplex (GQ). 5,10,15,20-Tetrakis(*N*-methyl-4-pyridinium)porphyrin (H_2_TMPyP) is one of the most studied porphyrins in this field, and although reported as presenting high affinity to GQ, its poor selectivity for GQ over duplex structures is recognized. To increase the desired selectivity, porphyrin modifications either at the peripheral positions or at the inner core through the coordination with different metals have been handled. Herein, studies involving the interactions of TMPyP and analogs with different DNA sequences able to form GQ and duplex structures using different experimental conditions and approaches are reviewed. Some considerations concerning the structural diversity and recognition modes of G-quadruplexes will be presented first to facilitate the comprehension of the studies reviewed. Additionally, considering the diversity of experimental conditions reported, we decided to complement this review with a screening where the behavior of H_2_TMPyP and of some of the reviewed metal complexes were evaluated under the same experimental conditions and using the same DNA sequences. In this comparison under unified conditions, we also evaluated, for the first time, the behavior of the Ag^II^ complex of H_2_TMPyP. In general, all derivatives showed good affinity for GQ DNA structures with binding constants in the range of 10^6^–10^7^ M^−1^ and ligand-GQ stoichiometric ratios of 3:1 and 4:1. A promising pattern of selectivity was also identified for the new Ag^II^ derivative.

## 1. Introduction

The discovery of telomeres was reported first by H. Muller in 1938 and soon after, in 1941, by McClintock [1,2,3]. In these earliest studies, both authors showed that each chromosome end is limited by a structure called telomere. The main functions of this type of structures are: (i) to maintain the stability of the structure of the chromosomes; (ii) to ensure that the genetic information is perfectly copied when the cell duplicates and (iii) to prevent the end junction between consecutive chromosomes which can lead to deoxyribonucleic acid (DNA) degradation or genetic mutations and consequently to the appearance of tumors [2].

Further studies demonstrated that during the process of cell division, telomeres undergo shortening since interruptions in the DNA replication process occur. As a defense to the mechanism involved in telomeres shortening, the enzyme telomerase was identified [2,4]. Telomerase is an enzyme that consists of several components including an endogenous ribonucleic acid (RNA) template of eleven nucleotides and a reverse transcriptase that adds specific and repetitive DNA sequences to the 3′ end of the chromosomes, preventing their shortening [2]. A requisite for telomerase activity is the existence of the single-strand DNA, to which the RNA template is connected by complementarity and then allows telomerase to perform its function of telomeric repeat addition (Figure 1).

The proposal of the enzyme telomerase as a potential target for anticancer drug design results from the evidence that this enzyme is overexpressed in almost 90% of human cancers [5,6].

Other important targets for anti-cancer drug design are human oncogenes and tumor suppressor genes due to their close association with the appearance of cancer cells. Oncogenes can result from the mutation of proto-oncogenes, the genes responsible for normal cell division, stimulation and death. Therefore, oncogenes involved in the initiation and progression of tumors have also been recognized as targets for the development of new anticancer drugs [7,8]. An interesting discussion concerning the challenges and potential advantages of targeting telomeric G-quadruplex (GQ) compared to gene promoter G-quadruplexes and to protein or enzyme targets was reported by Balasubramanian et al. [7].

The presence of repetitive sequences of the nucleobases thymine (T), adenine (A) and guanine (G), in the sequence TTAGGG (abbreviated as T_2_AG_3_), and of secondary structures, such as the G-quadruplexes (GQ) at the end of telomeres (whose function is to protect telomere ends from nuclease attack) enables indirect targeting of telomerase [9,10,11]. GQ are higher-order DNA structures formed by the self-assembly of four guanine (G) bases in a planar quadrangular arrangement via Hoogsteen hydrogen bonding, known as G-quartets (Figure 2). The subsequent stacking of these quartets on top of each other via *π-π* interactions can give rise to different GQ conformations, as is exemplified in Figure 2 for an intermolecular (bimolecular) GQ conformation [12].

The stability of the GQ structures depends on the presence of cations, whose location results from a balance between cation repulsion and attractive interactions with oxygen atoms from carbonyl groups.

Ions such as NH_4_^+^ and K^+^ with ionic radii (i.r.) of 1.43 Å and 1.33 Å, respectively, are too bulky to coordinate within the plane of a G-quartet, thus their coordination occurs with eight oxygen atoms between two stacked G-quartets (Figure 3A). In the case of cations with a small ionic radius, such as Na^+^ (i.r. 0.95 Å), the coordination with four oxygen atoms within the plane of a single quartet is possible (Figure 3B).

Depending on the number of DNA strands involved in the GQ arrangement, conformations mediated by intra- or intermolecular interactions are formed. Structures formed from one, two or four separate strands of DNA give rise, respectively, to GQ in uni-, bi- or tetramolecular conformations [13,14]. The GQ structures are also dependent on the spatial orientation of the strands and are designated as parallel, antiparallel or hybrid depending on the relative strand orientation (Figure 4A). The formation and stability of all unimolecular (intramolecular) and bimolecular (intermolecular) GQ structures imply the presence of three or four loops with different sizes and sequences. The single-strand sections that are not involved in the G-quartet arrangement form loops that link the guanines bases. These loops can adopt different geometries, namely edgewise (or lateral), double-chain-reversal (or propeller) and diagonal (Figure 4B). The loop residues can further stabilize GQ structures through hydrogen bonds and by stacking interactions.

The relative strand orientation also influence the glycosidic bond angle of the guanines involved in the quartets that can assume an *anti* and/or *syn* geometry (Figure 5). A combination of *syn* and *anti* geometries is observed for antiparallel and hybrid structures, while parallel topologies contain almost exclusively *anti* geometry (Figure 5) [15,16].

All the described structural properties of GQ concerning not only the length, base composition and directionality of the loops, but also the size of the stabilizing metal cations, contribute to the existence of four grooves surrounded by the guanine phosphodiester backbone [17]; these cavities are similar to the minor and major grooves formed in the well-known double-stranded structure. In the GQ grooves, large variations in the widths, depths, inter-phosphate distances between the DNA strands and base orientations are also found (Figure 5).

Changes in strand direction also affect the guanine glycosidic torsion angles and will further alter the relative position of the sugar ribose and the groove dimension. In the case of parallel GQ, all the guanine glycosidic torsion angles are characteristic of an equivalent *anti* conformation, thus the four grooves’ dimensions are all equivalent. If one of the strand orientations changes to an antiparallel arrangement, changes in guanine glycosidic torsion angles occur and adopt both *syn* and *anti* conformations. These changes will further alter the groove dimensions, generating narrow, medium and wide grooves (Figure 5).

In the case of human telomeric DNA, the tandem repeats until the sequence T_2_AG_3_ form unimolecular intramolecular G-quadruplexes in the chromosomic end regions.

The recognition that GQ structures can be easily accessible in physiological conditions in the presence of monovalent cations such as NH_4_^+^, Na^+^ and K^+^ [19] was of particular value in the wide range of studies involving the investigation of GQ as targets for drug design [13,20,21,22].

It has been found that G-quadruplexes are recognized and partially unwound by telomerase for 3′-end extension, thus the binding of stabilizing compounds to G-quadruplex structures will “lock” the telomeres in the G-quadruplex configuration, preventing telomere lengthening by telomerase (Figure 6) [23].

Telomerase inhibitors should present higher selectivity for GQ DNA structures when compared with duplex DNA as the drug must be able to recognize GQ DNA in the presence of a large amount of duplex DNA, in the cellular nucleus. The interaction of the ligand with duplex will reduce its availability to bind GQ structures, resulting in a reduction of its telomerase inhibitory function [24].

The importance of developing ligands with adequate structural features to selectively stabilize GQ DNA structures in the presence of duplex DNA structures prompted us to discuss some important achievements concerning this topic.

Special attention will be given to the interactions of the positively charged ligand 5,10,15,20-tetrakis(*N*-methyl-4-pyridinium)porphyrin (H_2_TMPyP) with several DNA sequences able to form GQ and duplex structures. The impact of H_2_TMPyP complexation with different metal ions (e.g., Zn^II^, Co^III^, Ni^II^, Cu^II^, Pd^II^, Au^III^ and Mn^III^) in the ligand selectivity towards GQ structures will be also considered and, when adequate, the reference to other studies concerning analogues to H_2_TMPyP will also be presented.

In the selection of this issue, we have our interest in the synthesis, functionalization and characterization of porphyrins and analogues for different biological/medical applications [25,26,27,28,29], namely as DNA stabilizing agents and telomerase inhibitors [30,31,32,33,34,35,36,37].

## 2. Interactive G-Quadruplex Ligands and GQ Recognition Modes

In 1997, Sun et al. reported [38], for the first time, the inhibition of telomerase by interactive G-quadruplex ligands and, based on the results, anticipated the potential of the approach as a novel research line for anticancer drug design. Since then, the search for GQ interactive ligands with high specificity and affinity for GQ has become a central issue for different research groups [9,12,18,22,23,38,39].

A large number of small ligands have demonstrated ability to bind non-covalently to DNA and RNA GQ structures [9]. Most of these ligands contain several fused or non-fused aromatic rings or are aromatic macrocycles. An interesting overview of the literature highlighting structure–quadruplex interaction relationships of organic modular GQ ligands was published by Alexandra Paulo et al. [40].

Different GQ recognition modes are possible: intercalation (Figure 7A), outside stacking on the ends of the G-quartet core, also known as end-stacking (Figure 7B), and interaction with the backbone (core and loop bases) known as groove or loop binding (Figure 7C,D). Since the GQ structure is rigid and stable, the distortion triggered by the intercalation of a ligand has an energetic cost. In this way, outside stacking (Figure 7B–D) is seen as the more favorable binding mode [39].

Simple procedures using optical spectroscopic techniques like ultraviolet-visible absorption (UV-Vis), fluorescence and circular dichroism (CD) allow to evaluate the ligand affinity and selectivity to DNA structures, while other spectroscopic methods such as mass spectrometry (MS) and nuclear magnetic resonance (NMR) allow to obtain kinetic, thermodynamic, stoichiometric and conformational data in structure-activity relationship (SAR) studies [31,32,41,42,43]. The efficiency of several small molecules to inhibit telomerase by adduct formation with GQs has been studied using different methods with promising results [30,31,44,45,46,47,48].

Under the context of aromatic ligands containing fused rings, the dicationic anthraquinone derivative presented in Figure 8 was the first ligand reported to inhibit telomerase function [38]. Many other small ligands were then reported to inhibit telomerase activity with small half-maximal inhibitory concentration (IC_50_) values, like tri-substituted acridines (e.g., BRACO-19) [49], the perylene tetracarboxylic diimide derivative (PIPER) [50], the fluorinated polycyclic quinoacridinium cation (RHPS4) (Figure 8) and dibenzophenanthroline derivatives [51], among others [12,39,52].

Amongst the aromatic macrocycles, porphyrins, such as the cationic H_2_TMPyP and analogs appear as the most widely tested telomerase inhibitors [32,33,53,54,55].

Structural attributes such as the presence of an extended heteroaromatic moiety that can interact on the G-quartet surface of a G-quadruplex by *π−π* stacking, and flexible cationic charged terminal side chains are pointed as essential features for effective ligand binding to a quadruplex and optimal inhibition of telomerase activity [9]. Other important structural characteristics are the presence of bulky substituents to prevent intercalation with double-stranded DNA and good solubility in aqueous media. In general, aromatic/planar molecules such as porphyrins and analogs, especially if positively charged, present good GQ stabilizing properties with high affinity for G-quadruplexes (K_b_ ≥ 10^6^ M^−1^, where K_b_ corresponds to the binding constant) [56,57,58].

Ligands containing metals present higher interaction with G-quadruplexes due to stronger *π-π* interactions and/or coordination processes, thus being more selective for GQ [42,59]. A metal center can be predicted as a structural region that sets ligands in specific geometries, thus optimizing their binding properties. Ligands with different metal centers can assume different geometries. In addition to their structural characteristics, the presence of metal centers can reduce the electron density on coordinated aromatic ligands, increasing the strength of *π-π* interactions with G-quartets.

As mentioned above, in the following section special attention will be given to works where the tetracationic H_2_TMPyP and related porphyrins were key players and how their coordination with different metals affects the stabilization and selectivity for GQ over duplex. Moreover, the wide range of small polycyclic ligands studied in this field has been the subject of other excellent revisions [9,40,52].

### Porphyrins and Metalloporphyrins as Interactive G-Quadruplex Ligands

The cationic H_2_TMPyP is one of the most studied porphyrins in telomerase inhibition by adduct formation with telomeric GQ structures [38,56,60], and in the stabilization of non-telomeric biologically relevant GQs from human oncopromoters, bacterial genomes and viral genomes [8].

The structure of tetrapyrrolic macrocycles like porphyrins can be changed in the periphery, at the *meso* or *β-pyrrolic* positions, with several types of substituents or at the porphyrin inner core by metal coordination, affording stable metallo-based ligands [61,62].

The type of interaction between cationic porphyrins and DNA structures depends not only on the location, size and charge of peripheral substituents but also on the presence or absence of metal in the porphyrin core [14]. It has been reported that the presence, at *meso* positions, of pyridinium substituents not only favors interactions with DNA by *π-π* stacking but also improves its water solubility, thus being an important feature for telomerase inhibition [31,39,63,64].

Cationic porphyrins are able to interact with negatively charged phosphate groups of DNA structures, minimizing the electrostatic repulsions. Moreover, their diameter is compatible with the diameter of the central channel of GQ, thus interaction by end-stacking was described to be favorable [31].

As already mentioned, the H_2_TMPyP is one of the most studied porphyrins in this field, and although reported as presenting high affinity to GQ, its poor selectivity for GQ over duplex structures is recognized [60]. To increase its selectivity, porphyrin modifications, either at the peripheral positions or at the inner core, were envisaged. The complexation of porphyrins and analogs with metals appears to be a strategy to increase porphyrin selectivity.

Izbicka et al. [65] reported for the first time the ability of metalloporphyrins to interact with G-quadruplex structures present in the telomeric sequence. These studies involved cell-based biochemical assays and molecular modeling, namely using the In^III^ and Cu^II^ complexes of H_2_TMPyP and QP4 (Figure 9). The binding mode of porphyrins and metalloporphyrins was proposed to be *π-π* stacking, specifically end-stacking on the top of the G-quartets at the termini of the GQ structures. An enhanced binding affinity was proposed for metalloporphyrins through an additional electrostatic interaction induced by the presence of metal ions.

A study where the Mn^III^ metal complex of the TMPyP was activated with the potassium monopersulfate (KHSO_5_), an oxygen atom donor, forming a very reactive high-valent porphyrin Mn(V)=O species was reported by Vialas et al. [66]. The authors reported that efficient oxidative cleavage of the quadruplex could be mediated by this oxo-metalloporphyrin. Using Polyacrylamide Gel Electrophoresis analysis (PAGE), the authors were able to identify the location of damage; the obtained results showed that the metalloporphyrin was able to bind to the last G-quartet of the quadruplex structure via an external interaction. It has been found that the high-valent oxo-metalloporphyrin was also able to mediate both electron-abstraction or H-abstraction on guanine or thymine residues, respectively, within the GQ target.

Later, Shi et al. [67] published a scientific paper revisiting the telomerase inhibiting activity of H_2_TMPyP (or H_2_TMPyP_4_) and of a wide range of analogs such as TMPyP_3_, TMPyP_2_, QP_4_ and QP_3_, (Figure 9) among others. The authors were able to SAR rules related to the importance of the presence of positively charged substituents, its position at *meso* or *beta* positions and the influence of bulky substituents on the interaction with GQ. The percentage of telomerase inhibition by several complexes of H_2_TMPyP, namely the Zn^II^, Co^III^, Fe^III^, Ni^II^, Mn^III^, Cu^II^, Mg^II^, Pt^II^ and Pd^II^ ones, was also accessed using a primer extension assay. The authors found that the square planar Cu^II^ complex and the pyramidal Zn^II^ were the better inhibitors and correlated this with the unhindered face for stacking offered by these two metalloporphyrins.

The interaction of the complex Cu^II^TMPyP with tetramolecular GQ with general sequence T_4_G*_n_*T_4_, (*n* = 4 and 8) was studied by Keating et al. [68] using the spectroscopic methods UV-Vis, fluorescence, CD and electron paramagnetic resonance (EPR). A binding stoichiometric ratio of 2:1 (Cu^II^TMPyP-GQ) was described and end-stacking of Cu^II^TMPyP at each end of the G-quadruplex was proposed. Similar behavior has been described for the same porphyrin complex Cu^II^TMPyP and the T_4_G_n_T_4_ sequence, with *n* between 4 and 10 [69].

The Cu^II^, Ni^II^, Zn^II^ and Co^III^TMPyP derivatives were also explored in a study established to understand the effect of the coordinated metal, its geometry and of an additional positive charge (in the case of the Co^III^) on the interactions of these metalloporphyrins with human telomeric GQ [70]. This study points to the existence of two binding modes, consistent with the coexistence of end-stacking and intercalation in the case of Ni^II^ and Cu^II^ porphyrins in a stoichiometry of 4:1. For Zn^II^ and Co^III^, the presence of axial ligands justifies the interaction occurring exclusively by end-stacking in a 2:1 stoichiometric ratio.

In 2005, Dixon et al. published a paper describing how the change in the metal coordinated in the porphyrin core influences the kinetics and its mode of interaction [71]. Considering that, under physiological conditions, the nickel porphyrins are inert concerning redox processes and are not photoactivable, it is expected that they interact with the telomeres passively by stacking processes. On the other hand, the manganese derivatives should be able to interact with telomeres and damage them by oxidative processes within cells, but stacking interactions are impossible for Mn^III^ derivatives due to the presence of water as axial ligands.

Taking these facts into account, the binding properties of the Ni^II^ and Mn^III^ metalloporphyrins owning one of the *meso* substituents with different length and bulky elements (compounds 2 and 3, Figure 10) were studied by surface plasmon resonance (SPR), and the capacity of the Ni^II^TMPyP to inhibit telomerase was also evaluated by a telomeric repeat amplification protocol (TRAP) assay. It has been found that the nature of the metal influences not only the kinetics but also the ligand binding mode. Higher selectivity, a tenfold preference for quadruplex over duplex, was observed for Mn^III^TMPyP [71]. The authors also showed that the kinetics of drug interaction with GQ DNA seems to depend on the mode of binding—end-stacking vs. external binding—in the GQ grooves.

The kinetic constants for the association and dissociation of different porphyrins with duplex and quadruplex DNA were obtained and depending on the kinetic mode (slow or fast) different types of interaction were proposed.

An aromatic moiety that can interact, by stacking, with DNA is present in the porphyrins that presented fast kinetics. The Mn^III^ porphyrin derivatives (compounds 1, 2 and 4, Figure 10) showed slow kinetics that were explained either by the perturbation of the coordination sphere of the metal ion upon interaction with DNA, or, more likely, by a binding mode that is different from stacking (external binding).

Later, the same research group published a communication reporting the ability of a Mn^III^ pentacationic porphyrin to discriminate, by four orders of magnitude, the quadruplex from the duplex structures, probably as a result of combining in its structure a central aromatic core and four flexible cationic arms [72]. The study was performed using SPR to measure the noncovalent equilibrium binding constants and the TRAP assay. The authors suggested that the bulky cationic substituents surrounding the aromatic core of the ligand could be responsible for its poor affinity for duplex DNA. The hypothesis of the occurrence of interaction by stacking with the last tetrad of quadruplex DNA by loss of one axial ligand or fitting of an axial water ligand within the central ion channel was also presented.

Another study involving Ni^II^TMPyP, Mn^III^TMPyP, Co^III^TMPyP and Au^III^TMPyP was later described [60]. H_2_TMPyP analogs with bulky substituents were also studied. From the CD, fluorescence, SPR, telomerase assay and in vitro experiments, the authors revealed that the porphyrins with Mn^III^ and Co^III^ present lower stabilization properties towards G_3_(TTAG_3_)_3_ when compared to the free base, Ni^II^ and Au^III^ porphyrins (Figure 11).

The presence of water molecules on either side of the porphyrin core of Mn^III^TMPyP and Co^III^TMPyP (Figure 11B) was pointed out as a factor that prevents the porphyrin intercalation between the base pairs of DNA and, consequently, decreases the porphyrins binding affinity to double-stranded DNA. On the other hand, the authors revealed that axial water molecules might hinder the initial stacking of the porphyrin molecule at the G-quadruplex end. The in vitro cellular tests performed with free-base and Mn^III^TMPyP analogs containing bulky substituents showed that these ligands can penetrate cells and mediate some of the typical cellular effects of small GQ ligands.

A porphyrin scaffold, the 5,10,15,20-tetrakis(4-(1-methylpyridinium-2-yl)phenyl)-porphyrin (H_2_TMPy_2_PP, Figure 12) and its metal complexes with Ni^II^, Co^III^ and Mn^III^ were studied, aiming to evaluate the impact of the variation of the charge position and axial coordination in the center of the porphyrin in the binding affinity of the selected complexes. The MTMPy_2_PP derivatives were studied in the presence of GQ and duplex DNA structures using fluorescence resonance energy transfer (FRET), CD, SPR and NMR methods [73]. The authors conclude that the interaction with GQ of all the derivatives, even of the Co^III^ complex coordinated with two water molecules, occurs by a *π*-stacking-like mode with an external G-quartet. Later, Dejeu et al. [74] extended the previous study to the Ni^II^, Co^III^ and Mn^III^ complexes of the 5,10,15,20-tetrakis(4-guanidinophenyl)porphyrin (MTGP) (Figure 12).

The results obtained for these two series of metalloporphyrins based on the ligands MTGP and MTMPy_2_PP were then compared with those reported for the corresponding H_2_TMPyP. The authors noted that, when compared to the metalated MTMPyP porphyrin series, it is striking that the influence of the metal is completely different. From this study, the authors conclude that the MTGP and MTMPy_2_PP porphyrins were better GQ ligands than their MTMPyP counterparts and, on the contrary to the H_2_TMPyP metal derivatives, the metal had no influence on the observed dissociation constant (K_D_) values.

Zn^II^TMPyP was also explored in an interesting work where its interaction with a quadruplex structure stabilized by the unusual presence of lead ion (Pb-GQ) (Figure 13) was evaluated using CD and UV-Vis spectroscopy and mass spectrometry [75]. The Pb-GQ structure was found to be formed in a 1:1 stoichiometry (Pb^II^-GQ). The Zn^II^TMPyP was described as a Pb-GQ structure-stabilizing ligand. The steric hindrance of the axial ligand of Zn^II^ and the relatively rigid structure of Pb-GQ was pointed out as factors that precluded the ligand intercalation and an interaction exclusively by end-stacking was proposed.

Aiming to understand the mechanism of ligand-assisted GQ folding, the potential of H_2_TMPyP, and of its Zn^II^, Cu^II^ and Pt^II^ complexes, to induce GQ folding from the single-stranded sequence (TAGGG)_2_ in a buffer containing K^+^ was investigated [76]. The authors demonstrated, using CD and UV-Vis experiments, that only Zn^II^TMPyP was able to induce the folding of GQ structure of the studied sequence. A stoichiometry of 2:1 (Zn^II^TMPyP-[(TAGGG)_2_]_2_ GQ) by end-stacking with an affinity constant (K_a_) of about 10^6^ M^−1^ was also reported from UV-Vis and isothermal calorimetry (ITC) titrations. The order for the GQ stabilizing ability of the studied compounds was Zn^II^TMPyP ~ H_2_TMPyP > Cu^II^TMPyP > Pt^II^TMPyP. Pt^II^TMPyP was referred to as not owning GQ stabilizing properties.

Later, the interaction of Pt^II^TMPyP and Pd^II^TMPyP complexes with quadruplex structures present in telomeres and oncogene promotors was again described in a study performed by Sabharwal et al. [77] using UV-Vis, fluorescence and CD spectroscopies, FRET melting assays and resonance light scattering. The obtained results suggest that both porphyrin complexes interact with telomeric quadruplex by *π-π* stacking with a binding affinity of 10^6^–10^7^ M^−1^. A modest selectivity for quadruplex vs. duplex was described for both metalloporphyrins. Interesting results about the aggregation of Pt^II^TMPyP under porphyrin excess conditions using Tel22 as a template were found. The authors noted the dissolution of the aggregates at concentration ratios [Pt^II^TMPyP]/[Tel22] ≤ 2, reaching their maximum size at [Pt^II^TMPyP]/[Tel22]~8.

The Zn^II^TMPyP and Cu^II^TMPyP complexes were revisited in a study where the interactions between the H_2_TMPyP (or H_2_TMPyP_4_) and the isomeric structure H_2_TMPyP2 were evaluated with the tetramolecular sequences T_4_G_4_ and T_4_G_4_T in the presence of buffer solutions containing K^+^ or Na^+^ ions [54]. Using CD, UV-Vis and fluorescence spectroscopy, the authors elucidate the effect of the 3′-T on the stabilization of porphyrins, binding modes, affinities and stoichiometries. This study provides information about the influence of metal center substitution and modulation of peripheral groups on porphyrin binding to GQ structures and identifies Zn^II^TMPyP as a promising GQ ligand, its binding being once again proposed to occur by end-stacking.

The interaction of the Zn^II^TMPyP with three different GQ structures, (TG_4_T_4_)_4_, (G_4_T_4_G_4_)_2_ and AG_3_(T_2_AG_3_)_3_ with tetramolecular, bimolecular and unimolecular topologies, respectively, was also evaluated using transient absorption spectroscopy to monitor the triplet decay dynamics of Zn^II^TMPyP [78]. The coexistence of different binding modes via *π-π* stacking of porphyrin macrocycle and the G-quartets was quantitatively identified and described as being intercalation/end-stacking for (G_4_T_4_G_4_)_2_ and AG_3_(T_2_AG_3_)_3_ and end-stacking/partial intercalation for (TG_4_T_4_)_4_. The authors reinforced that the intercalation process is undesirably affected by the steric hindrance of the axial water. Binding stoichiometric ratios of 1:2 for (TG_4_T_4_)_4_ and AG_3_(T_2_AG_3_)_3_, and of 1:1 for (G_4_T_4_G_4_)_2_, were reported for GQ/Zn^II^TMPyP adducts.

The GQ stabilization ability and selectivity of the gold(III) porphyrin Au^III^TMPyP was recently re-evaluated using biophysical and biochemical assay [79,80] and compared with the behavior of other porphyrin derivatives (Figure 11A) already studied by Romera et al. [60]; porphyrins were also tested as inhibitors of telomerase. The authors showed an increase in the binding affinity of the porphyrin to the GQ target when a Au^III^ ion was present in the porphyrin core. Modeling studies suggested that the insertion of the square planar Au^III^ ion favors *π-π* staking with stronger electrostatic interactions, since an extra positive charge is added to the porphyrin and a decrease in the electron density is induced.

An overview of the main obtained data with the cationic porphyrins selected in the studies reported here, like the DNA sequences used, binding mode, affinity, porphyrin:GQ stoichiometry, experimental conditions and the most relevant techniques, can be found below in Table 1. The structures of the porphyrins referred in Table 1 are resumed in Figure 14.

The analysis of a high number of scientific articles comprising the interaction of H_2_TMPyP, its metal complexes and some analogues with different DNA sequences by recurring to different methodologies and experimental conditions motivated us to complement this review with a screening where the behavior of H_2_TMPyP and of its Zn^II^, Co^III^, Ni^II^, Cu^II^, Pd^II^ and Mn^III^ complexes as interactive GQ ligands was evaluated under the same experimental conditions and using the same GQ sequences. It is important to highlight that this evaluation allows comparing different complexes under unified conditions, facilitating the clear identification of the most promising compound. Taking advantage of the spectroscopic features of the porphyrins and metalloporphyrins, the possible impact of the experimental conditions on the interaction mode of these ligands was evaluated using readily accessible techniques such as UV-Vis and fluorescence spectroscopy. The screening was extended for the first time to the Ag^II^ complex of H_2_TMPyP, and the selectivity of the ligands for GQ vs. duplex DNA structures was evaluated by performing the studies also in the presence of Salmon Sperm DNA.

## 3. Evaluation of the Interactions H_2_TMPyP and of Its Metal Complexes (M = Ag^II^, Zn^II^, Co^III^, Ni^II^, Pd^II^, Mn^III^ and Cu^II^) with GQ and ds DNA Structures

### 3.1. UV-Vis Spectroscopy

The free-base H_2_TMPyP and the corresponding Zn^II^, Ni^II^, Cu^II^, Ag^II^, Pd^II^, Mn^III^ and Co^III^ metallo complexes were prepared according to literature procedures [25,81]. In the selection of the complexes, we had into account several studies reporting that the Zn^II^ and Cu^II^ complexes are the better GQ stabilizers and telomerase inhibitors. The controversy concerning the influence of the axial water in the interaction of the Mn^III^ or Co^III^ porphyrin complexes prompted us also to evaluate the behavior of these complexes. The Ag^II^ complex of TMPyP was considered since, to the best of our knowledge, this is the first time that this complex was evaluated as a GQ stabilizer.

The studies were undertaken using three different DNA sequences which give rise to different GQ topologies (Table 2 and Figure 15), namely the tetramolecular sequence (TG_4_T)_4_, the bimolecular *Oxytricha* repeat oligonucleotide (G_4_T_4_G_4_)_2_ [82] and the human telomeric repeat (AG_3_(T_2_AG_3_)_3_) [83]. A long-chain DNA strand (salmon sperm) was also used to compare the affinity and to evaluate the selectivity of the tested porphyrins for G-Quadruplex structures.

The oligonucleotide (TG_4_T_4_)_4_ assumes a parallel tetramolecular topology with guanine residues with glycosidic torsion angles in *anti* geometry, forming a right-handed helical structure with four equivalent grooves. This GQ is a *Tetrahymena* telomeric repeat sequence that results from the arrangement of four separate strands, thus not including any loop.

The oligonucleotide (G_4_T_4_G_4_)_2_ presents four stacked G-quartets and two groups of four thymine residues involved in the formation of two diagonals loops, thus acquiring a symmetrical antiparallel bimolecular GQ topology. Adjacent strands are alternately in parallel and antiparallel orientations, the guanine residues being consecutively in *syn* and *anti* geometries, in agreement with the *syn*-*syn*-*anti*-*anti* of the glycosidic torsion angles around each G-quartet. The same topology in the presence of K^+^ and Na^+^ ions was found in NMR solution studies [82].

The human telomeric sequence AG_3_(T_2_AG_3_)_3_ could adopt distinct topologies in K^+^ buffer solutions, which are different from the ones reported in the presence of Na^+^ cation [83]. The crystal structure of the K^+^ form reveals that all GGG segments are in parallel positions with guanine residues in the *anti* geometry [84]. TTA loops connect the top and the bottom of two GGG strands and are in a double chain reversal conformation. The loop residues are positioned next to the grooves rather than at the quadruplex ends. The presence of these reversal loops limits the access to the grooves and so, the external quartets expose their planar surface areas that become available to aromatic ligands binding. The human telomeric sequence is also described as a mixture of antiparallel and hybrid structures [15]. Structure and topology of the three studied GQ are represented in Figure 15. In the case of the unimolecular GQ, the most common topologies are presented.

The typical electronic absorption spectra of porphyrins with a highly intense Soret band ranging from 413 to 461 nm accompanied by less intense Q bands (four for the free-bases and two for metalloporphyrins) between 500 and 650 nm, prompted us to use UV-Vis to obtain qualitative and quantitative information about GQ/porphyrin interactions [41,42].

Under this context, when a ligand interacts with DNA structures, changes in its characteristic absorbance bands, such as hypochromic/hyperchromic and bathochromic (typically red shift) alterations, can occur depending on the type of interactions. The bathochromic shift is associated with a decrease in the *π*-*π** transition energy due to the coupling of the *π* bonding orbital of the DNA base pairs with the empty *π** antibonding orbital of the ligand [32].

As a result of an intercalative binding process, typical values of hypochromicity higher than 35% and bathochromicity (red shift) above 15 nm in the Soret band are expected; it is important to take into account that these values were determined for long pieces of duplex DNA where the end-stacking is not significant [32,67,85]. Due to the less direct contact between *π*-systems, changes in the UV-Vis absorption spectra are less remarkable for groove binding or outside binding, for which red shifts lesser than 8 nm have been described [86,87]. Thus, by analyzing the batho- and hypochromic effects on the obtained spectra at the end of the titrations, it is possible to evaluate the affinity, the selectivity and to predict the type of interaction.

The UV-Vis titrations of the silver complex Ag^II^TMPyP, the free-base H_2_TMPyP and of the other MTMPyP complexes (M = Zn^II^, Co^III^, Ni^II^, Pd^II^, Mn^III^ and Cu^II^) were performed by adding aliquots of the selected GQ topologies, referred above in Table 2, and of the double-stranded DNA structure in phosphate buffer saline (PBS) to each ligand; the titrations were stopped after obtaining a constant absorbance during three successive additions of each oligonucleotide (see details of the experimental procedures and of the structure, molar extinction coefficient (ε) and absorbance maximum (λ_max_) of all the studied porphyrins in Appendix A). Blank experiments were performed with PBS buffer before the titration of each ligand with the selected DNA structures (data not shown).

The UV-Vis spectra obtained with H_2_TMPyP and with the metal complexes MTMPyP, where M = Ag^II^, Zn^II^, Co^III^, Ni^II^, Pd^II^, Cu^II^ and Mn^III^, at the end of the titrations with each GQ structures are summarized in the Appendix A. The results of red shifts, hypochromism percentage, stoichiometry (L:DNA) and K_b_ values obtained for H_2_TMPyP and for the metal complexes in the presence of each DNA structure are summarized in Table 3.

As an example, the changes observed in the UV-Vis spectra of Ag^II^TMPyP during its titration with the different GQ structures AG_3_(T_2_AG_3_)_3_ (unimolecular), G_4_T_4_G_4_ (bimolecular) and TG_4_T (tetramolecular) and with the salmon sperm double-stranded DNA are presented in Figure 16 (see also Appendix A).

From the data obtained it is possible to observe that the Ag^II^ derivative presents high affinity and selectivity for all the GQ structures, since red shifts between 9 and 13 nm and binding constants (K_b_) in the range of 1.63 × 10^6^–1.13 × 10^7^ (Table 3, entry 1) were found. In the case of double-stranded (ds) salmon sperm, a red shift of 7 nm and a Kb of 6.97×10^4^ was observed.

The observed red shifts and hypochromism percentages between 18% and 22%, observed in the titrations with GQ, are consistent with an external interaction, probably by end-stacking [86,87]. The high difference observed between the unimolecular GQ, AG_3_(T_2_AG_3_)_3,_ (K_b_ = 1.13 × 10^7^) and the ds DNA (K_b_ = 6.97 × 10^4^) structures is highly indicative of the selectivity of the Ag^II^TMPyP ligand towards GQ structures. The binding constant observed for Ag^II^TMPyP is 10-fold higher than the one observed for the free-base H_2_TMPyP (Table 3, entry 2) and it is the highest value obtained for the metalloporphyrins (Table 3, entries 3–7). These results, along with the observed selective behavior, indicate that Ag^II^TMPyP can be a promising GQ stabilizing ligand. The 3:1 ligand-to-GQ (L:GQ) stoichiometry obtained for the tetra and bimolecular topologies and 4:1 L:GQ for the unimolecular one reinforces the ability of this ligand to interact with GQ structures.

The differences (lower red shift and hypochromic effect) observed between the bimolecular and the GQ in tetra and unimolecular conformations can be justified by the presence of two diagonal loops in the bimolecular conformations (Figure 15) that can difficult the ligand binding to the GQ structure in special if end-stacking is considered.

For the Mn^III^ complex, the interaction with both the GQ tetramolecular and the double-stranded salmon sperm was almost absent (Appendix A; this lack of interaction was not introduced in Table 3). These results confirm the previous reports, [60,71,72] pointing to the poor stabilization and stacking interactions by Mn^III^ complexes due to the presence of water molecules as axial ligands; no more studies were performed with this complex.

Overall, from the data obtained it is possible to observe that, with exception of the Mn^III^ complex, the free-base H_2_TMPyP and the metalloporphyrins present good affinity for both DNA structures. Higher affinity for the GQ structures, when compared to the observed for double-stranded DNA, was observed when considering the unimolecular conformation AG_3_(T_2_AG_3_)_3_, for which the K_b_ are in the range of 10^6^–10^7^ M^−^^1^.

For Co^III^TMPyP (Table 3, entry 4 and Appendix A) the most significant bathochromic effect was observed for the unimolecular GQ (with a red-shift of 12 nm) while the bimolecular GQ structure showed a blue shift of 3 nm; the insignificant or absent bathochromic effect observed in the spectra of the tetramolecular conformation (Δλ = 0 nm) and of the salmon sperm (Δλ = 1 nm) can be associated with the presence of water molecules as axial ligands confirming their negative influence in the macrocycle interaction with DNA structures, as previously reported [71,72].

For the Ni^II^TMPyP derivative (Table 3, entry 5), a K_b_ with 10^6^ M^−1^ order was found for the GQ and double-stranded structures, pointing to the low selectivity of this derivative for GQ structures. This low selectivity for GQ structures was also observed for the widely studied H_2_TMPyP. In general, when compared with the free base, the Ni^II^ derivative showed higher K_b_ values.

The Pd^II^TMPyP (Table 3, entry 6) complex showed high affinity for the GQ structures, especially for the unimolecular topology, and a pattern of selectivity was also identified, especially when comparing unimolecular GQ and the double-stranded structure.

Similar behavior was observed for the Cu^II^TMPyP (Table 3, entry 7) ligand, although the obtained data points this complex to having lower affinity for DNA structures, which is consistent with their lower binding constants when compared to the data here reported for the other studied complexes, except for Co^III^TMPyP (Figure 17).

Red shifts lower than 15 nm and hypochromic percentages, in general, lower than 35% were observed for all the studied complexes, except for the Ni^II^ complex that showed hypochromic percentages around 40%, pointing to the occurrence of interaction by external stacking of ligands in the GQ structure. The type of interaction of the free-base H_2_TMPyP has been involved in controversy in the scientific community, with some authors pointing to intercalation as the binding mode, while a higher number of authors point to interaction as occurring via external binding [31,60,88].

It is interesting to note that, with exception of the Ag^II^TMPyP derivative, when looking to the obtained K_b_ values, the interaction of the MTMPyP derivatives with the studied bimolecular GQ, (G_4_T_4_G_4_)_2_, is again weaker. As mentioned before, this fact could be related to the presence, in the GQ structure, of two diagonal loops that can limit the access of the ligands to the terminal tetrads. The high affinity of the Ag^II^TMPyP for the bimolecular GQ is evidence of the high affinity of this derivative for GQ structures.

For better visualization of the impact of the ligands in each GQ and double-stranded DNA, the K_b_ observed at the end of the UV-Vis titration of the MTMPyP ligands and the selected DNA sequences are compared in Figure 17A,B.

When analyzing the obtained binding constants, K_b_ values in the range of 2.5 × 10^5^−1.6 × 10^7^ M^−1^ were obtained in general for the GQ structures, pointing to high affinity for these structures; for the studied double-stranded DNA, K_b_ values in the range of 6.7 × 10^4^–2.6 × 10^6^ M^−1^ were found, representing slight interactions and confirming ligands selectivity for GQ structures. The results obtained for the Ag^II^TMPyP complex are very important since they point to this ligand as a promising ligand for selective GQ stabilization, since the obtained K_b_ values are higher for GQ and much lower (21- or 5-fold) for the ds DNA than those obtained for the H_2_TMPyP or the Zn^II^TMPyP derivatives, both described as promising GQ stabilizing agents.

In general, the obtained results (Table 3) are in good agreement with the previous ones reported in literature (Table 1). The order of magnitude found for the K_b_ of free-base H_2_TMPyP with the tetramolecular structure (TG_4_T)_4_ is in accordance with the one reported by Boschi et al. for a similar tetramolecular GQ sequence (10^6^ M^−1^). In addition, the observed hypochromism and red-shift values are close to the reported values.

Our results suggest that the introduction of Zn(II) increased the binding affinity of the free-base porphyrin to all tested GQ sequences, just as observed by Boschi et al. Still, the experimental K_b_ found for the unimolecular GQ AG_3_(T_2_AG_3_)_3_ (4.33 × 10^6^ M^−1^) was higher than the one reported by DuPont et al. (7.6 × 10^5^ M^−1^) using different techniques. The K_b_ found for Co^III^TMPyP with AG_3_(T_2_AG_3_)_3_ (1.00 × 10^6^ M^−1^) was one order of magnitude superior to the one described by DuPont et al. (1.2 × 10^5^ M^−1^), while for Ni^II^TMPyP it was one order of magnitude lower (4.8 × 10^6^ vs. 7.4 × 10^7^ M^−1^, respectively). For Pd^II^TMPyP, a significant resemblance was observed between our experimental results (9.3 × 10^6^ M^−1^) and the value reported by Sabarwal et al., (1.0 × 10^7^ M^−1^), for the AG_3_(T_2_AG_3_)_3_. The K_b_ value of Cu^II^TMPyP with the unimolecular GQ (2.3 × 10^6^ M^−1^) is significantly inferior to the one described by DuPont et al., (1.7 × 10^10^ M^−1^). Nonetheless, the red shift and hypochormism values retrieved from the titrations with the tetramolecular GQ are in proximity to the ones reported by Boschi et al., for similar GQ sequences.

### 3.2. Fluorescence Experiments

The G-Quadruplex fluorescent intercalator displacement (G4-FID) assay is another well-established method to evaluate and confirm the affinity of a ligand for GQ and in particular its selectivity for GQ. This assay is based on the loss of fluorescence of a probe, the thiazole orange (TO), as a result of its displacement from DNA by a ligand [41,42,45]. The concentration of the ligands required to decrease the fluorescence of the probe (TO) by 50% is noted by DC_50_.

To validate the UV-Vis data obtained for the Ag^II^ complex, its ability to displace TO from the unimolecular GQ structure AG_3_(T_2_AG_3_)_3_ and from the double-stranded salmon sperm was evaluated by fluorescence spectroscopy (Figure 18).

As can be seen, the results obtained for the Ag^II^TMPyP, red line for the displacement of TO from the GQ structure (DC_50_ = 0.99 uM) and green line for the displacement of TO from the double-stranded structure (DC_50_ = 2.03 uM), point to a pattern of selectivity already highlighted by the UV-Vis data. For comparison, the results with the H_2_TMPyP and the Zn^II^TMPyP are also present, confirming the lack of selectivity of the free-base porphyrin. Similar behavior of selectivity was observed for both the Zn^II^ and the Ag^II^ metal complexes.

## 4. Conclusions

Overall, the reported data show that the free-base H_2_TMPyP, analogues and metalloporphyrin counterparts present good affinity for the GQ DNA structures, with the exceptions of the Mn^III^ and Co^III^ complexes, for which axial ligands are present. In general, the authors reinforced that the intercalation process is undesirably affected by the steric hindrance of the axial water in metal complexes, the external stacking being pointed as the most probable binding mode.

The binding mode of porphyrins and metalloporphyrins occurs mainly by *π-π* stacking, specifically end-stacking on the top of the G-quartets at the termini of the quadruplex structures. An enhanced binding affinity was proposed for metalloporphyrins through an additional electrostatic interaction induced by the presence of metal ions. It has been found that the nature of the metal influences not only the kinetics but also the ligand binding mode.

Complexes containing square planar (Cu^II^) and pyramidal (Zn^II^) geometries revealed to be the better inhibitors and their geometry was correlated with the unhindered face for stacking offered by porphyrins owning these metal centers. In the case of the Cu^II^TMPyP complex, a binding stoichiometric ratio of 2:1 (Cu^II^TMPyP-GQ) was described, and end-stacking of Cu^II^TMPyP in both G-quadruplex ends was proposed.

In the case of the Mn^III^ porphyrin derivatives, slow kinetics were described. Perturbation of the coordination sphere of the metal ion upon interaction with DNA or a binding mode that is different from stacking (external binding) were pointed to explain this behavior. The presence of water molecules on its porphyrin core was indicated as a factor that prevents the porphyrin intercalation between the base pairs of DNA and, consequently, decreases the porphyrins binding affinity to double-stranded DNA. On the other hand, the authors recognize that axial water molecules might hinder the initial stacking of the porphyrin molecule on the G-quadruplex end, thus justifying their lower stabilization properties. Similar behavior was described in the case of the Co^III^ complex.

The order for the GQ stabilizing ability was Zn^II^TMPyP~H_2_TMPyP > Cu^II^TMPyP > Pt^II^TMPyP. Pt^II^TMPyP was referred to as not owning GQ stabilizing properties. Modeling studies suggested that the insertion of the square planar Au^III^ ion favors *π-π* staking with stronger electrostatic interactions derived from the presence of an extra positive charge in the porphyrin core that induces a decrease in the electron density and an increase in the binding affinity of the porphyrin to the GQ target.

The collected data in the spectroscopic screening, performed under unified conditions, suggests that the free-base H_2_TMPyP and the metalloporphyrins present good affinity for both DNA structures, with the exceptions of the Mn^III^ and Co^III^ complexes. In particular, higher affinity constants for the GQ structures, when compared to those observed for double-stranded DNA, were obtained, especially for the unimolecular conformation AG_3_(T_2_AG_3_)_3_, for which the K_b_ values are in the range of 10^6^–10^7^ M^−1^. In the case of the Mn^III^ and Co^III^ complexes, the lower stabilization properties, probably resulting from the presence of water molecules as axial ligands, already observed by other authors, were confirmed.

The new derivative Ag^II^TMPyP showed good affinity for GQ-DNA structures, with binding constants in the range of 10^6^–10^7^ M^−1^ and ligand-GQ stoichiometric ratios of 3:1 and 4:1.

The process of discovering ligands with high affinity and selectivity for GQ is challenging and could open new horizons on the anticancer therapies based on the detection and stabilization of G-quadruplexes. Considering the affinity of the Ag^II^ complex for GQ structures and a predictable pattern of selectivity, demonstrated out by the UV-Vis and fluorescence screening presented herein, further studies must be performed to confirm and complement the obtained data.

## Figures and Tables

**Figure 1 biomolecules-11-01404-f001:**
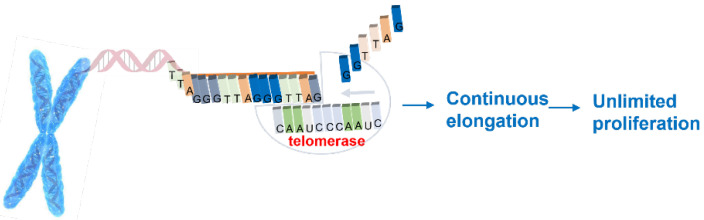
Telomerase function.

**Figure 2 biomolecules-11-01404-f002:**
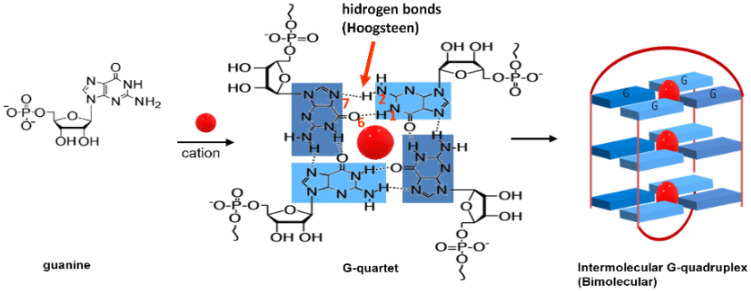
G-quartet and a possible G-quadruplex structure.

**Figure 3 biomolecules-11-01404-f003:**
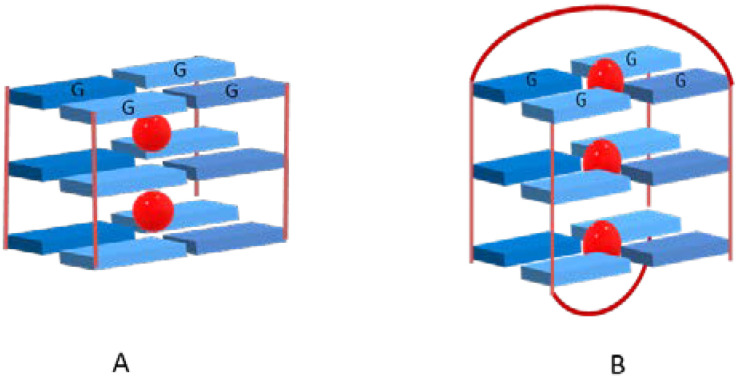
Cation positions between stacked quartets (**A**) or within the plane (**B**) in a G-quadruplex containing three quartets (for simplicity, backbones were omitted).

**Figure 4 biomolecules-11-01404-f004:**
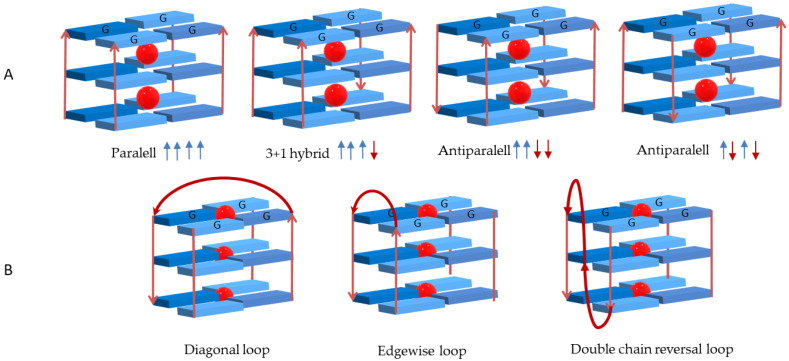
(**A**) Different conformations assumed by G-quadruplexes structures depending on strand orientation and (**B**) possible loop geometries.

**Figure 5 biomolecules-11-01404-f005:**
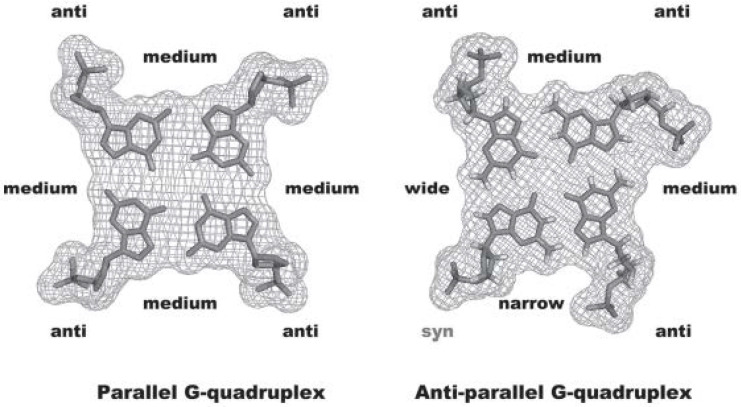
*Anti* and *syn* conformation of glycosidic torsion angles and groove sizes of parallel and anti-parallel GQ structures, (figure reused from [18], with permission from John Wiley and Sons.

**Figure 6 biomolecules-11-01404-f006:**
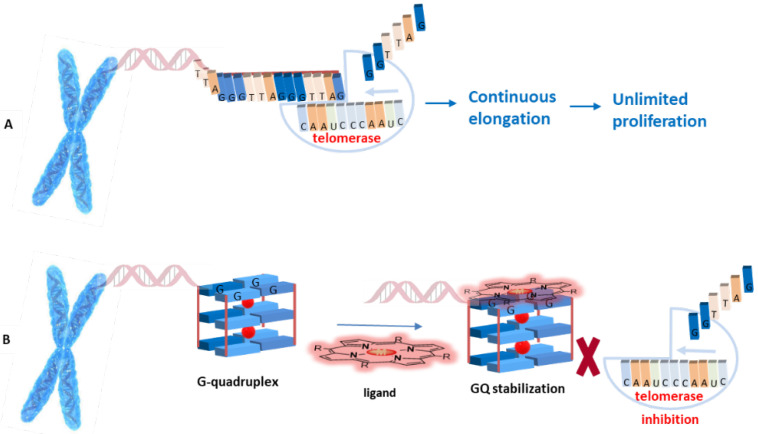
(**A**) Telomere end elongation by telomerase; (**B**) G-quadruplex-ligand adduct formation for indirect telomerase inhibition.

**Figure 7 biomolecules-11-01404-f007:**
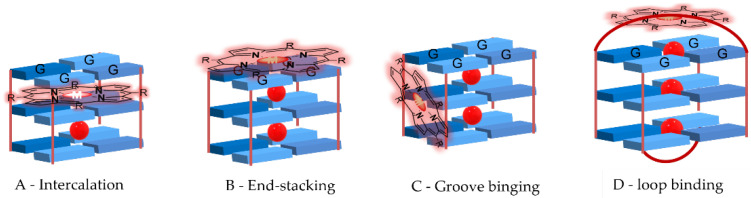
(**A**–**D**) Types of interactions ligand-G-quadruplexes. For simplicity, grooves are not represented.

**Figure 8 biomolecules-11-01404-f008:**
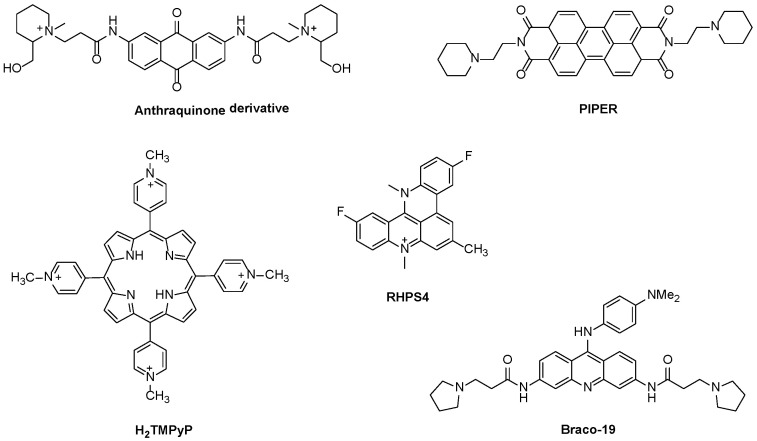
Examples of polycyclic ligands reported to inhibit telomerase activity.

**Figure 9 biomolecules-11-01404-f009:**
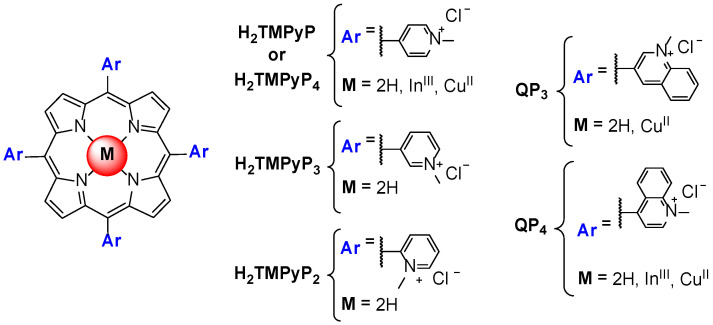
Structure of the first porphyrins and metalloporphyrins evaluated as GQ ligands.

**Figure 10 biomolecules-11-01404-f010:**
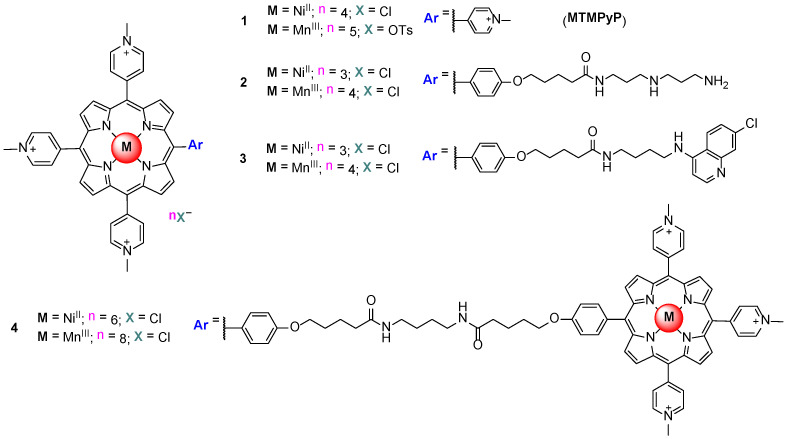
Structure of the Ni^II^ and Mn^III^ porphyrins derivatives studied by Dixon et al. [71].

**Figure 11 biomolecules-11-01404-f011:**
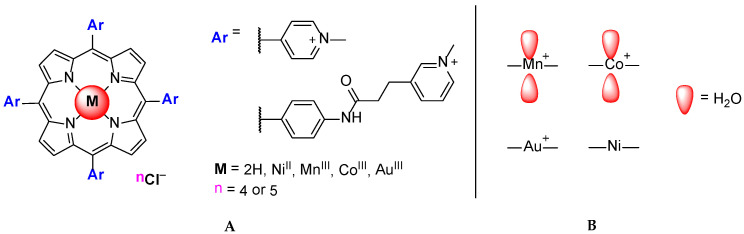
Structure of H_2_TMPyP and analogs with bulky substituents (**A**); presence or absence of axial substituents in different metal ions (**B**) studied by Romera et al. [60].

**Figure 12 biomolecules-11-01404-f012:**
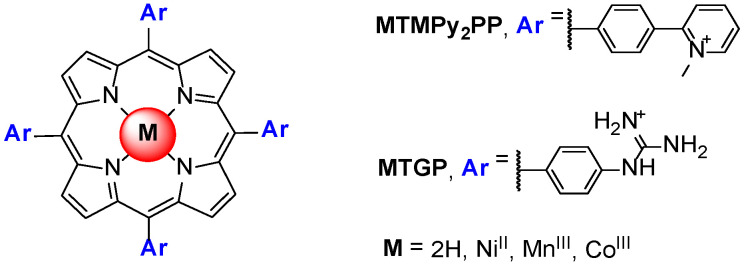
Structures of 5,10,15,20-tetrakis(4-(*N*-methyl-pyridinium-2-yl)phenyl)-porphyrin (MTMPy_2_PP) and 5,10,15,20-tetrakis(4-guanidinophenyl)porphyrin (TGP) derivatives, studied by Sabater et al. [73].

**Figure 13 biomolecules-11-01404-f013:**
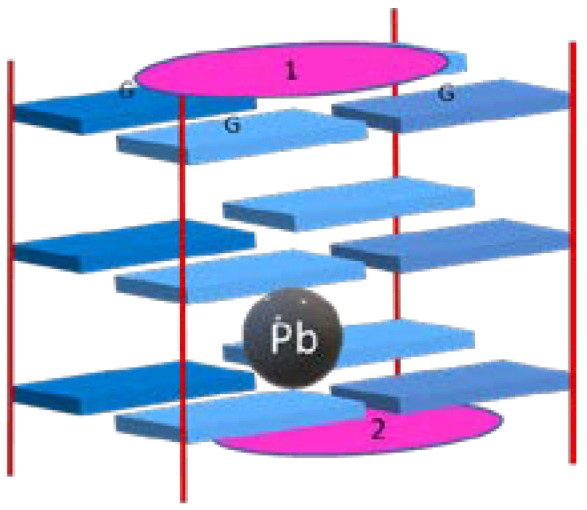
Pb-GQ structure with end-stacking of two Zn^II^TMPyP molecules (pink ellipses).

**Figure 14 biomolecules-11-01404-f014:**
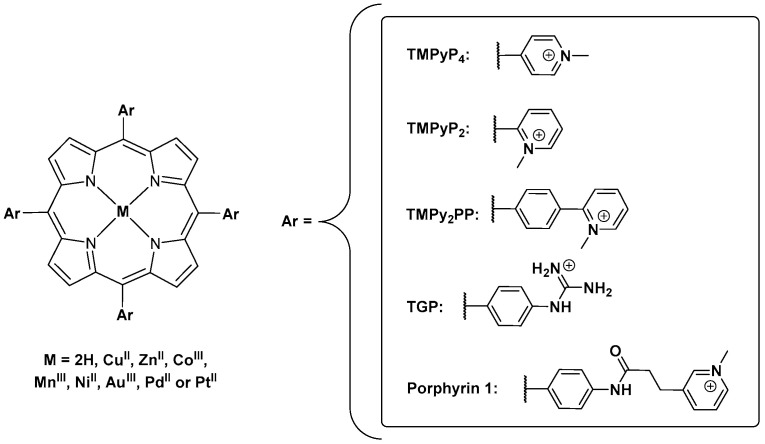
Structures of the porphyrins referred in Table 1.

**Figure 15 biomolecules-11-01404-f015:**
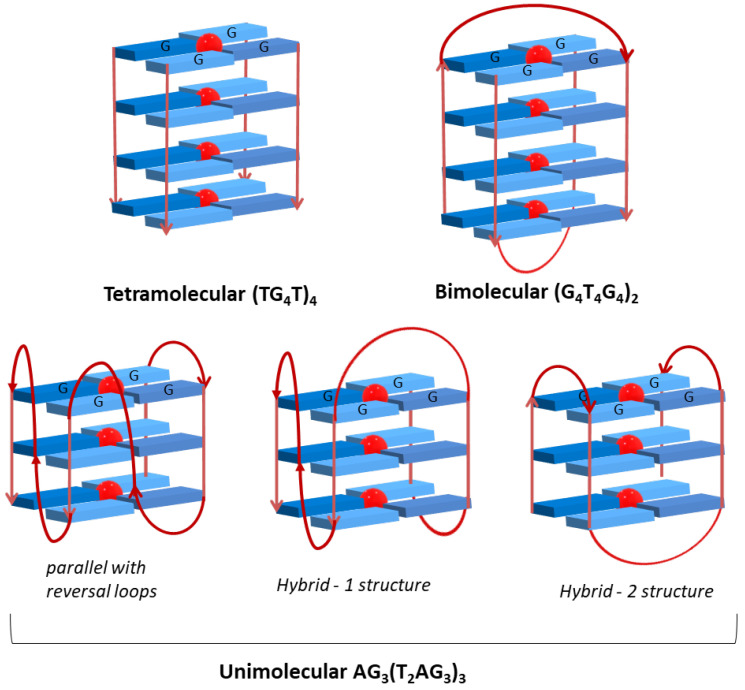
Structure and possible topology of the studied G-quadruplexes.

**Figure 16 biomolecules-11-01404-f016:**
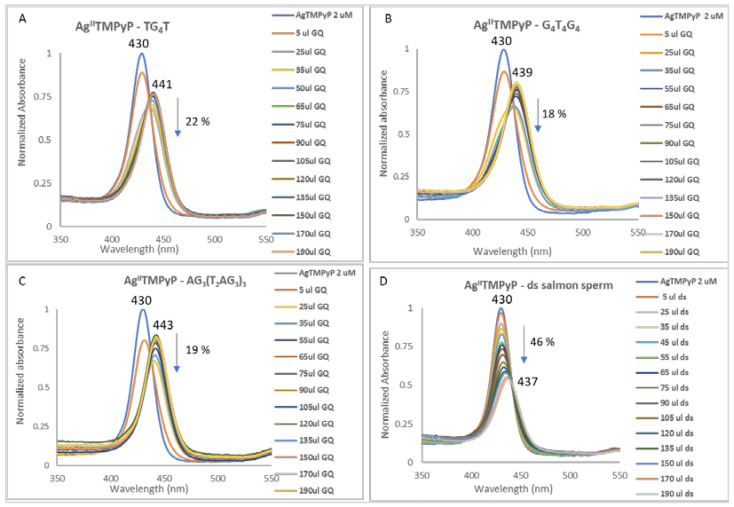
UV-Vis spectra (350–650 nm) obtained for Ag^II^TMPyP and (**A**) tetramolecular GQ, (**B**) bimolecular GQ, (**C**) unimolecular GQ and (**D**) salmon sperm double-stranded DNA. DNA structures were prepared in 20 mM PBS buffer with 100 mM KCl.

**Figure 17 biomolecules-11-01404-f017:**
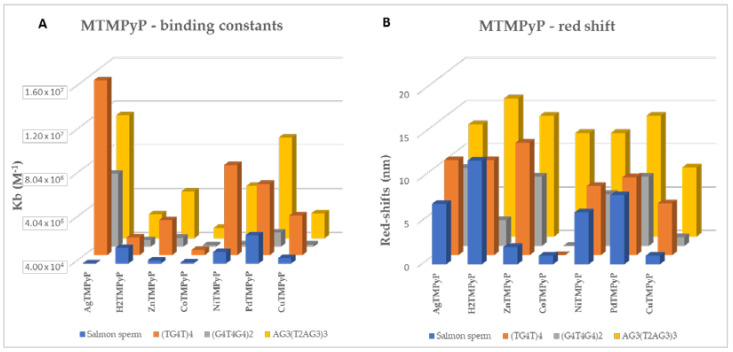
(**A**) binding constants (K_b_) (M^−1^) and (**B**) red shift observed at the end of the UV-Vis titration of the MTMPyP ligands and the selected DNA sequences.

**Figure 18 biomolecules-11-01404-f018:**
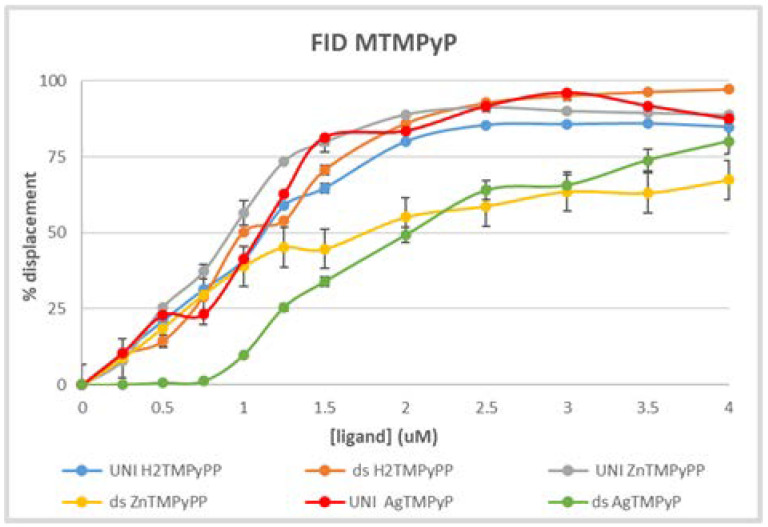
G4-FID assay performed for the in PBS at 25 °C with the Ag^II^TMPyP using the unimolecular GQ (AG_3_(T_2_AG_3_)_3_) and double-stranded salmon sperm. A comparison with the results obtained with the H_2_TMPyP and the Zn^II^TMPyP is presented.

**Table 1 biomolecules-11-01404-t001:** Compilation of results obtained for the MTMPyP and analogues (vide *infra* Figure 14) by different selected authors (organized by alphabetic order), using different methodologies.

Reference	Porphyrin *	Metal	DNA Sequence (5′-3′)	Major Findings				Buffer	Relevant Techniques
				Ka (M^−1^) or K_D_	Binding Mode	Porphyrin: GQ Stoichiometry	Other Parameters		
Boschi et al. [54]	TMPyP_4_	Free-base	(T_4_G_4_)_4_	K_a_ = (3.7 ± 0.9) × 10^6^	End-stacking and possibly intercalation	2:1	T_1/2_ = 312.6 ± 0.4 K; %H = 35 ± 1; Δλ = 13.4 ± 0.7 nm	KPI	UV-Vis, fluorescence, CD, melting
			(T_4_G_4_T)_4_	K_a_ = (3.0 ± 1.5) × 10^6^			T_1/2_ = 317.7 ± 1.0 K; %H = 36 ± 1; Δλ = 15.4 ± 0.3 nm		
		Cu(II)	(T_4_G_4_)_4_	K_a_ = (1.4 ± 0.6) × 10^6^	End-stacking and possibly intercalation	1:1	T_1/2_ = 310.1 ± 0.4 K; %H = 22 ± 1; Δλ = 7.1 ± 0.7 nm		
			(T_4_G_4_T)_4_	K_a_ = (1.1 ± 0.7) × 10^6^			T_1/2_ = 315.7 ± 1.3 K; %H = 26 ± 2; Δλ = 8.3 ± 0.1 nm		
		Zn(II)	(T_4_G_4_)_4_	K_a_ = (16 ± 6.0) × 10^6^	End-stacking and groove- or outside-binding	2:1	T_1/2_ = 324.6 ± 0.6 K; %H = 33 ± 3; Δλ = 11.5 ± 0.2 nm		
			(T_4_G_4_T)_4_	K_a_ = (11 ± 2.0) × 10^6^			T_1/2_ = 336.2 ± 0.6 K; %H = 32 ± 1; Δλ = 14.2 ± 0.3 nm		
	TMPyP_2_	Free-base	(T_4_G_4_)_4_	K_a_ = (2.0 ± 1.0) × 10^6^		1:1	T_1/2_ = 320.9 ± 0.3 K; %H = 23 ± 4%; Δλ = 5.7 ± 0.6 nm		
			(T_4_G_4_T)_4_	K_a_ = (6.3 ± 0.6) × 10^6^	End-stacking and groove- or outside-binding	1:1	T_1/2_ = 329.8 ± 0.3 K; %H = 22 ± 2%; Δλ = 6.3 ± 0.6 nm		

Dejeu et al. [74]	TMPyP_4_	Free-base	(T_2_AG_3_T)	K_D_ = 62 nM	End-stacking	not available	not available	HEPES	SPR, FRET
			(G_3_T_2_A)_3_G_3_T_2_	K_D_ = 345 nM					
			(CG_)4_T_4_(CG)_4_ (duplex)	K_D_ = 345 nM					
		Co(II)	(T_2_AG_3_T)	K_D_ = 6600 nM					
			(G_3_T_2_A)_3_G_3_T_2_	K_D_ = 5500 nM					
			(CG_)4_T_4_(CG)_4_ (duplex)	n.b.					
		Mn(III)	(T_2_AG_3_T)	K_D_ = 9000 nM					
			(G_3_T_2_A)_3_G_3_T_2_	K_D_ = 10,000 nM					
			(CG_)4_T_4_(CG)_4_ (duplex)	n.b.	n.b.				
		Ni(II)	(T_2_AG_3_T)	K_D_ = 59 nM	End-stacking	not available			
			(G_3_T_2_A)_3_G_3_T_2_	K_D_ = 240 nM					
			(CG_)4_T_4_(CG)_4_ (duplex)	K_D_ = 200 nM					
	TMPyP_2_	Free-base	(T_2_AG_3_T)	K_D_ = 52 nM					
			(G_3_T_2_A)_3_G_3_T_2_	K_D_ = 114 nM					
			(CG_)4_T_4_(CG)_4_ (duplex)	K_D_ = 404 nM					
		Co(II)	(T_2_AG_3_T)	K_D_ = 3.4 nM					
			(G_3_T_2_A)_3_G_3_T_2_	K_D_ = 15 nM					
			(CG_)4_T_4_(CG)_4_ (duplex)	K_D_ = 417 nM					
		Mn(III)	(T_2_AG_3_T)	K_D_ = 4 nM			not available		
			(G_3_T_2_A)_3_G_3_T_2_	K_D_ = 29 nM					
			(CG_)4_T_4_(CG)_4_ (duplex)	n.b.	n.b.				
		Ni(II)	(T_2_AG_3_T)	K_D_ = 29 nM	End-stacking				
			(G_3_T_2_A)_3_G_3_T_2_	K_D_ = 5.4 nM					
			(CG_)4_T_4_(CG)_4_ (duplex)	K_D_ = 185 nM					
	TGP	Free-base	(T_2_AG_3_T)	K_D_ = 83 nM					
			(G_3_T_2_A)_3_G_3_T_2_	K_D_ = 290 nM					
			(CG_)4_T_4_(CG)_4_ (duplex)	K_D_ = 16,000 nM					
		Mn(III)	(T_2_AG_3_T)	K_D_ = 20 nM					
			(G_3_T_2_A)_3_G_3_T_2_	K_D_ = 24 nM					
			(CG_)4_T_4_(CG)_4_ (duplex)	n.b.	n.b.				
		Ni(II)	(T_2_AG_3_T)	K_D_ = 18 nM	End-stacking	not available			
			(G_3_T_2_A)_3_G_3_T_2_	K_D_ = 53 nM					
			(CG_)4_T_4_(CG)_4_ (duplex)	K_D_ = 1030 nM					

DuPont et al. [70]	TMPyP_4_	Co(III)	AG_3_(T_2_AG_3_)_3_	K_a_ = (1.2 ± 0.7) = × 10^5^	End-stacking	2:1	ΔH_1_ = −2.8 ± 0.1 kcal/mol; −TΔS_1_ = −4.1 ± 0.1 kcal/mol	K^+^ BPES	ICT, CD, ESI/MS
		Ni(II)	AG_3_(T_2_AG_3_)_3_	K_a_ = 7.4 ± 0.8) × 10^7^	End-stacking and intercalation	4:1	ΔH_1_ = −3.2 ± 0.3 kcal/mol; −TΔS_1_ = −7.6 ± 0.4 kcal/mol		
		Cu(II)	AG_3_(T_2_AG_3_)_3_	K_a_ = (1.7 ± 1.1) × 10^10^			ΔH_1_ = −4.2 ± 0.1 kcal/mol; −TΔS_1_ = −9.2 ± 0.6 kcal/mol		
		Zn(II)	AG_3_(T_2_AG_3_)_3_	K_a_ = (7.6 ± 0.6) × 10^5^	End-stacking	2:1	ΔH_1_ = −4.6 ± 0.4 kcal/mol; −TΔS_1_ = −3.4 ± 0.4 kcal/mol		

Keating et al. [68]	TMPyP_4_	Cu(II)	(T_4_G_4_T_4_)_4_	K_a_ = 5.6 × 10^6^	End-stacking	2:1	%H = 50; Δλ = 9 nm	KPi	UV-Vis, fluorescence, CD, EPR
			(T_4_G_8_T_4_)_4_	K_a_ = 5.2 × 10^7^	End-stacking and intercalation	3:1	%H = 58; Δλ = 12 nm		

Romera et al. [60]	TMPyP_4_	Free-base	(T_2_AG_3_T)	K_a_ = 1.6 × 10^7^	End-stacking	not available	not available	HEPES	SPR, FRET, CD, TRAP assay
			(G_3_T_2_A)_3_G_3_T_2_	K_a_ = 2.9 × 10^6^					
			(CG_)4_T_4_(CG)_4_ (duplex)	K_a_ = 2.9 × 10^6^					
		Au(III)	(T_2_AG_3_T)	K_a_ = 2.2 × 10^6^					
			(G_3_T_2_A)_3_G_3_T_2_	K_a_ = 1.2 × 10^6^					
			(CG_)4_T_4_(CG)_4_ (duplex)	K_a_ = 4.3 × 10^6^					
		Co(III)	(T_2_AG_3_T)	K_a_ = 1.5 × 10^5^	End-stacking	not available	not available		
			(G_3_T_2_A)_3_G_3_T_2_	K_a_ = 1.8 × 10^5^					
		Mn(III)	(T_2_AG_3_T)	K_a_ = 1.1 × 10^5^	End-stacking	not available	not available	HEPES	
			(G_3_T_2_A)_3_G_3_T_2_	K_a_ = 1.0 × 10^5^					
		Ni(II)	(T_2_AG_3_T)	K_a_ = 1.7 × 10^7^					
			(G_3_T_2_A)_3_G_3_T_2_	K_a_ = 4.2 × 10^6^					
			(CG_)4_T_4_(CG)_4_ (duplex)	K_a_ = 5.0 × 10^6^					
	Porphyrin 1	Mn(II)	(T_2_AG_3_T)	K_a_ = 1.7 × 10^7^					
			(G_3_T_2_A)_3_G_3_T_2_	K_a_ = 1.8 × 10^7^					
			(CG_)4_T_4_(CG)_4_ (duplex)	n.b.	n.b.				

Sabater et al. [73]	MA	Co(III)	(T_2_AG_3_T)	K_D_ = (17 ± 0.4) nM	End-stacking	not available	not available	HEPES	FRET melting assay, SPR, CD, NMR
			(G_3_T_2_A)_3_G_3_T_2_	K_D_ = (60.2 ± 1.9) nM					
			(CG_)4_T_4_(CG)_4_ (duplex)	K_D_ = (3660 ± 13.4) nM					

Sabharwal et al. [77]	TMPyP_4_	Pd(II)	AG_3_(T_2_AG_3_)_3_	K_a_ = (1.0 ± 0.3) × 10^7^	End-stacking	6.5:1	ΔT = 30.9 ± 0.4 °C;	KPi	UV-Vis, fluorescence and CD spectroscopies, FRET melting assays, and resonance light scattering
		Pt(II)	AG_3_(T_2_AG_3_)_3_	K_a_ = (5.8 ± 0.8) × 10^6^		7:1	ΔT = 30.7 ± 06 °C		

Abbreviations: K_a_ = affinity (or binding) constant; K_D_ = dissociation constant; n.b. = non-binding; *T*_1/2_ = temperature of half transition; %H = hipochromic percentage; Δλ = red shift; ΔT—melting temperature deviation * Porphyrin structures showed in Figure 14.

**Table 2 biomolecules-11-01404-t002:** Sequence and topology of studied oligonucleotides.

Oligonucleotide Sequence	Topology	Abbreviation
5′-TGGGGT-3′ (*Tetrahymena* telomeric repeat)	Tetramolecular G-Quadruplex	(TG4T)_4_
5′-GGG GTT TT GGG G-3′ (*Oxytricha* repeat oligonucleotide)	Bimolecular G-Quadruplex	(G_4_T_4_G_4_)_2_
5′-AGG GTT AGG GTTAGG GTT AGGG-3′ (human telomeric repeat)	Unimolecular G-Quadruplex	AG_3_(T_2_AG_3_)_3_
long single strand	Double-strand DNA	Salmon-sperm DNA

**Table 3 biomolecules-11-01404-t003:** Red shift, % hypochromism and Kb values obtained for H_2_TMPyP and its metal complexes.

		G-Quadruplexes (GQ)			Double-Stranded (ds)	
Entry		(TG_4_T)_4_	(G_4_T_4_G_4_)_2_	AG_3_ (T_2_AG_3_)_3_	Salmon Sperm	Ligand (L)
	Red shift (nm)	11	9	13	7	**Ag^II^TMPyP**
(1)	% Hypochromism	22	18	19	46	
	L:DNA Stoichiometry	3:1	3:1	4:1	n.a.	
	K_b_ (M^−1^)	(1.63 ± 0.31) × 10^6^	(6.72 ± 0.41) × 10^6^	(1.13 ± 0.41) × 10^7^	(6.97 ± 0.37) × 10^4^	
	Red shift (nm)	11	3	16	12	**H_2_TMPyP**
(2)	% Hypochromism	23	41	28	36	
	L:DNA Stoichiometry	3:1	3:2	4:1	1:1	
	K_b_ (M^−1^)	(1.66 ± 0.41) × 10^6^	(6.43 ± 0.44) × 10^5^	(2.57 ± 0.47) × 10^6^	(1.49 ± 0.32) × 10^6^	
	Red shift (nm)	13	8	14	2	**Zn^II^TMPyP**
(3)	% Hypochromism	17	74.6	27	5	
	L:DNA Stoichiometry	3:1	3:1	4:1	2:1	
	K_b_ (M^−1^)	(3.25 ± 0.38) × 10^6^	(8.57 ± 0.76) × 10^5^	(4.33 ± 0.43) × 10^6^	(3.52 ± 0.74) × 10^5^	
	Red shift (nm)	0	−3	12	1	**Co^III^TMPyP**
(4)	% Hypochromism	26	23	32	49	
	L:DNA Stoichiometry	3:1	3:1	4:1	1:1	
	K_b_ (M^−1^)	(5.30 ± 0.89) × 10^5^	(1.86 ± 0.58) × 10^5^	(1.00 ± 0.39) × 10^6^	(1.36 ± 0.40) × 10^5^	
	Red shift (nm)	8	6	12	6	**Ni^II^TMPyP**
(5)	% Hypochromism	40	42	35	24	
	L:DNA Stoichiometry	3:1	3:1	3:1	1:1	
	K_b_ (M^−1^)	(8.28 ± 0.39) × 10^6^	(2.56 ± 0.41) × 10^6^	(4.84 ± 0.44) × 10^6^	(1.11 ± 0.39) × 10^6^	
	Red shift (nm)	9	8	14	8	**Pd^II^TMPyP**
(6)	% Hypochromism	5	12	14	24	
	L:DNA Stoichiometry	4:1	3:1	4:1	2:1	
	K_b_ (M^−1^)	(6.55 ± 0.38) × 10^6^	(1.35 ± 0.41) × 10^6^	(9.26 ± 0.36) × 10^6^	(2.64 ± 0.44) × 10^6^	
	Red shift (nm)	6	1	8	1	**Cu^II^TMPyP**
(7)	% Hypochromism	28	15	28	3	
	L:DNA Stoichiometry	4:1	3:1	3:1	1:1	
	K_b_ (M^−1^)	(3.67 ± 0.42) × 10^6^	(2.49 ± 0.41) × 10^5^	(2.33 ± 0.44) × 10^6^	(5.58 ± 0.79) × 10^5^	

n.a.—not available; DNA—GQ or ds.

## Data Availability

The experimental data presented in this study are available in Appendix A.

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
