# Peer review of "The Interactions of H2TMPyP, Analogues and Its Metal Complexes with DNA G-Quadruplexes—An Overview"

_biomolecules, 2021, doi:10.3390/biom11101404_

Round 1

Reviewer 1 Report

The manuscript reports a spectroscopic study aimed at elucidating the selectivity of binding of a library of metalloporphyrins to different GQ structures, especially selectivity compared to the binding of double strand DNA. The paper is well written and the results support the claims of the authors. 

Of all the metalloporphyrins chosen for this study, only AgTMPyP was never studied as GQ stabiliser. The discussion does not offer much in terms of comparison of the results obtained in this study with literature reports. Could the authors expand this? 

The authors confirm the strongest binding of AgTMPyP to GQ by fluorescence spectroscopy, i.e., via the displacement of a dye from GQ and double helix: would it be possible for the authors to add error bars to the points that appear on the graph?

Author Response

The authors would like to thank the reviewer corrections and recommendations which certainly improved the quality of the paper.

Comment of the reviewer

Of all the metalloporphyrins chosen for this study, only AgTMPyP was never studied as GQ stabiliser. The discussion does not offer much in terms of comparison of the results obtained in this study with literature reports. Could the authors expand this? 

Our answer

We appreciate and thank the reviewer for this important comment. According with the reviewer suggestion, in the revised manuscript, a paragraph including a comparison of the results obtained with the results reported in literature was added.

 Comment of the reviewer

The authors confirm the strongest binding of AgTMPyP to GQ by fluorescence spectroscopy, i.e., via the displacement of a dye from GQ and double helix: would it be possible for the authors to add error bars to the points that appear on the graph?

Our answer

According with the reviewer suggestion, in the revised manuscript, error bars were added to the points that appear on the graph related with FID results.

Reviewer 2 Report

Review article by Ramos et al. presents data collected on on interaction of porphyrine analogues and metal complexes with G-quadruplexes. The overview of the field is thorough and explanatory in a way that makes it easy for reader to follow and understand concepts and ideas.

Comments:

  1. Most of the figures that are not adapted from cited literature are designed with the same schematic presentation of G-quadruplex structures. Some of the figures,  however (for example, Figure 13 and 15), are not. Using the same schematics for G-quadruplexes would add to the uniformity of the article and help reader unfamiliar with different topologies to follow more easily
  2. Several Figures that were adapted from literature display unedited borders. These should be removed (Figures 4, 5, 15).
  3. Figure 6B should use the same schematics; chromosome is missing on the right part. Type and colour of the text on the Figure as well as what it describes should be uniform, which is currently not.
  4.  Several typing mistakes can be found throughout the text, sentences should be either paraphrased or errors corrected (see for example line 134, 150, 156, 158, 197, 444, etc.)
  5. Term G-quadruplex and its abbreviation is introduced in the abstract, yet naming throughout the article is inconsistent and should be reviewed (quadruplex, G-quadruplex, G-Quadrupplex, GQ...). Using quadruplex on its own is erroneous - there are other known quadruplex structures that are not formed by guanine residues

Author Response

Response to Reviewer 2 Comments:

General Comment of the reviewer

Review article by Ramos et al. presents data collected on on interaction of porphyrine analogues and metal complexes with G-quadruplexes. The overview of the field is thorough and explanatory in a way that makes it easy for reader to follow and understand concepts and ideas.

Our answer

The authors thank the reviewer for the positive comment and for recognizing the quality of the review.

Comment of the reviewer

Most of the figures that are not adapted from cited literature are designed with the same schematic presentation of G-quadruplex structures. Some of the figures, however (for example, Figure 13 and 15), are not. Using the same schematics for G-quadruplexes would add to the uniformity of the article and help reader unfamiliar with different topologies to follow more easily

Our answer

The authors thank the reviewer for the comment. Figures 13 and 15 were changed and, in the revised version, and are presented in a similar schematic presentation as proposed by the reviewer.

Comment

Several Figures that were adapted from literature display unedited borders. These should be removed (Figures 4, 5, 15).

Our answer

The authors thank the reviewer for this alert. According with the suggestion the Figures were improved and the borders removed.

Comment

Figure 6B should use the same schematics; chromosome is missing on the right part. Type and colour of the text on the Figure as well as what it describes should be uniform, which is currently not.

Our answer

The authors thank the reviewer for the alert. Type and colour of the text on the figure was corrected accordingly.

 Comment

 Several typing mistakes can be found throughout the text, sentences should be either paraphrased or errors corrected (see for example line 134, 150, 156, 158, 197, 444, etc.)

Our answer

The authors thank the reviewer for this alert and the sentences and typos were corrected accordingly.

 Comment

Term G-quadruplex and its abbreviation is introduced in the abstract, yet naming throughout the article is inconsistent and should be reviewed (quadruplex, G-quadruplex, G-Quadrupplex, GQ...). Using quadruplex on its own is erroneous - there are other known quadruplex structures that are not formed by guanine residues

 Our answer

The authors thank the reviewer for the comment. The word quadruplex was changed to G-quadruplex or its abbreviation, in accordance to the reviewer comment.

Reviewer 3 Report

In this manuscript, Ramos et al summarized recent advances in the studies of porphyrin derivative H2TMPyP and its interactions with GQs. Analogs and metal complexes of H2TMPyP attract much interest due to their prospects as G4-stabilizing agents. Limited selectivity impedes widespread application of H2TMPyP. However, rational choice of metal cations and substituents allows to solve the selectivity issue. For that, detailed analysis of ligand-GQ and ligand-duplex binding modes was needed. The authors complemented the analysis of the relevant literature with their own experimental data. Those data were helpful for comparing different complexes under unified conditions and revealed the most promising compound clearly. Thus, the manuscript covers important issues and is technically sound. Structure of the manuscript could be improved. The major points are listed below:

  1. The manuscript is a mix of a review and a research paper. The choice of this hybrid article type should be justified. If it seems appropriate, please consider dividing the manuscript into a conventional review and a short communication/data in brief and perhaps submitting the two pieces in parallel.
  2. The authors begin the story of H2TMPyP with telomers, which is understandable, and do not mention other GQ targets, which is perhaps an omission. The title suggests a broad scope of the manuscript. Non-telomeric GQs are mentioned few times, but never discussed thoroughly. I recommend commenting early in Introduction that the ligand has also been tested for stabilizing non-telomeric biologically relevant GQs (e.g., those from human oncopromoters, bacterial genomes and viral genomes).
  3. Section 2.1 is poorly structured. I recommend subsections and subsections, e.g., ‘Porphyrins and metalloporphyrins: interactions with GQs and duplexes’; ‘(Metallo)porphyrins in complexes with GQs: stoichiometry and binding modes’, ‘Metalloporphyrins with GQs: the role of metal cation’, ‘Substituents in (metallo)porphyrins and axial ligands’, ‘Selectivity issue: approaches to improve specificity for GQs’.
  4. Table 1. It could be more informative. Is Kd indeed the only major finding? Could the authors incorporate data on stabilization, presumed binding mode, etc. in the table?
  5. Delignate clearly binding affinity from the stabilizing effect. In 3.1 the authors state ‘this is the first time that this complex was evaluated as GQ stabilizer’. That’s not true. I see no evaluation of  Ag-TMPyP as a GQ stabilizer in Section 3. UV-Vis and FID assays are fine, but to postulate effects on stability you need difference in the melting temperatures (or at least CD amplitudes).
  6. In 3.1. the rationale for the choice of targets (GQs or different topologies and dsDNA) is explained nicely. It implies that selectivity for particular GQ topologies (parallel/antiparallel/hybrid) can be verified in addition to GQ over dsDNA selectivity. Please comment.

Minor issues

  1. Table 1 - add errors for Kd values
  2. 16. Changes in the Q band cannot be seen. Instead of showing the full range spectrum with the Q band, please expand the Soret band.
  3. 17. Vertical axis should be labeled
  4. English editing or proof-reading is needed.

Author Response

General Comment of the reviewer

In this manuscript, Ramos et al summarized recent advances in the studies of porphyrin derivative H2TMPyP and its interactions with GQs. Analogs and metal complexes of H2TMPyP attract much interest due to their prospects as G4-stabilizing agents. Limited selectivity impedes widespread application of H2TMPyP. However, rational choice of metal cations and substituents allows to solve the selectivity issue. For that, detailed analysis of ligand-GQ and ligand-duplex binding modes was needed. The authors complemented the analysis of the relevant literature with their own experimental data. Those data were helpful for comparing different complexes under unified conditions and revealed the most promising compound clearly. Thus, the manuscript covers important issues and is technically sound. Structure of the manuscript could be improved.

Our answer

The authors thank the reviewer for the positive comments and pertinent suggestions to improve the paper quality and soundness.

 Comment of the reviewer

The manuscript is a mix of a review and a research paper. The choice of this hybrid article type should be justified. If it seems appropriate, please consider dividing the manuscript into a conventional review and a short communication/data in brief and perhaps submitting the two pieces in parallel.

Our answer

The authors thank the reviewer for the proposal. As pointed in the covering letter, the submission of the review in this uncommon form was conscious and discussed amongst the authors.

Actually, and according with the reviewer suggestion we tried previously to divide the document in two articles  but after the division we felt that each  article lost the desired soundness and impact. Under this context we opted to maintain the unusual hybrid composition (we found some reviews with this composition)   in order to maintain the scientific interest of the readers.  In addition, during this bibliographic review, we verify the absence of studies including the Ag complex, fact that encourage us to make and publish a screening using this complex.

Moreover, and as stated by the reviewer, “the authors complemented the analysis of the relevant literature with their own experimental data. Those data were helpful for comparing different complexes under unified conditions and revealed the most promising compound clearly. Thus, the manuscript covers important issues and is technically sound.” Although in the first submission we try to justify this hybrid form we hope in the revised submission this  justification  is now better supported.

 Comment of the reviewer

The authors begin the story of H2TMPyP with telomers, which is understandable, and do not mention other GQ targets, which is perhaps an omission. The title suggests a broad scope of the manuscript. Non-telomeric GQs are mentioned few times, but never discussed thoroughly. I recommend commenting early in Introduction that the ligand has also been tested for stabilizing non-telomeric biologically relevant GQs (e.g., those from human oncopromoters, bacterial genomes and viral genomes).

Our answer

The authors thank the reviewer for the suggestion. The use of H2TMPyP for stabilizing non-telomeric biologically relevant GQs (e.g., those from human oncopromoters, bacterial genomes and viral genomes)” is now referred in introduction and  it is supported by the addition of adequate references.

Comment of the reviewer

Section 2.1 is poorly structured. I recommend subsections and subsections, e.g., ‘Porphyrins and metalloporphyrins: interactions with GQs and duplexes’; ‘(Metallo)porphyrins in complexes with GQs: stoichiometry and binding modes’, ‘Metalloporphyrins with GQs: the role of metal cation’, ‘Substituents in (metallo)porphyrins and axial ligands’, ‘Selectivity issue: approaches to improve specificity for GQs’. 

 Our answer

The authors thank the suggestion, and we understand the reviewer point of view. However considering that the article was organised chronologically and that each reported article contains several TMPyP metal complexes, that includes (or not), for instance, metals with axial ligands, and presenting (or not) conclusions related with stoichiometries, it was very difficult for us to envisage section 2.1 separated in sub-sections without compromising the article fluency and logic. So, section 2.1 was not divided but it is important to mention that the new information added to Table 1 (according with the reviewer suggestion, see next answer) some of those aspects are now summarized on the referred Table and consequently clearer.

Comment of the reviewer

Table 1. It could be more informative. Is Kd indeed the only major finding? Could the authors incorporate data on stabilization, presumed binding mode, etc. in the table? 

Our answer

We appreciate and thank the reviewer for the comment. The constants were pointed as major findings considering the importance of this parameter (is present in almost all the reported papers) and is one of the parameters obtained from our spectroscopic screening. However, in the revised version, wherever possible, binding modes, stoichiometries, melting temperatures, etc. were added, according to the reviewer suggestion.

Comment of the reviewer

Delignate clearly binding affinity from the stabilizing effect. In 3.1 the authors state ‘this is the first time that this complex was evaluated as GQ stabilizer’. That’s not true. I see no evaluation of  Ag-TMPyP as a GQ stabilizer in Section 3. UV-Vis and FID assays are fine, but to postulate effects on stability you need difference in the melting temperatures (or at least CD amplitudes).

Our answer

We appreciate and thank the reviewer for the pertinent comment.

As mentioned above, during this bibliographic review, we verify the absence of studies including the Ag complex, fact that encourage us to make and publish a screening using this complex.

The words “points” and “suggests” were used when conclusions were taken. A sentence referring to further studies, using other complementary techniques, will be performed, was added to the conclusions.

Comment of the reviewer

In 3.1. the rationale for the choice of targets (GQs or different topologies and dsDNA) is explained nicely. It implies that selectivity for particular GQ topologies (parallel/antiparallel/hybrid) can be verified in addition to GQ over dsDNA selectivity. Please comment.

Our answer

We very much appreciate and thank the reviewer for the point of view concerning the possibility of evaluating the selectivity of ligands for different topologies depending on  strand orientation, parallel, antiparallel or mix.

Our main objective of studying the selected GQ topologies (tetra, bi and unimolecular) was to compare the different GQs, in terms of the hindrance to the ligand access caused by the presence of loops, in special the diagonal loops present in the bimolecular GQ that probably will difficult the end-stacking of ligands, usually described for porphyrins.

By the other side, the bimolecular quadruplex G4T4G4, is described as an antiparallel GQ. The other two selected GQ forms parallel structures. Thanking this in account the obtained results, suggest that these porphyrins could present selectivity for parallel GQ structures. However further studies should be performed, using complementary methods, to confirm the supposed selectivity pattern. In future this interesting point will surely be taken in account when new studies were planned.

Minor issues pointed by the reviewer

Table 1 - add errors for Kd values

Our answer

The authors thank for the advertisement and for the careful and accurate analysis of the paper. In the revised version of the paper the errors were added to table 1, according with the reviewer comment.

Minor issues pointed by the reviewer

  1. Changes in the Q band cannot be seen. Instead of showing the full range spectrum with the Q band, please expand the Soret band.

Our answer

The authors thank for the advertisement and for the careful and accurate analysis of the paper. In the revised version of the paper the spectra were expanded and are now in accordance with the reviewer suggestion.

Minor issues pointed by the reviewer

  1. Vertical axis should be labeled

Our answer

The authors thank for the advertisement and for the careful and accurate analysis of the paper. The vertical axis was added to graphic A.

Minor issues pointed by the reviewer

English editing or proof-reading is needed.

Our answer

The text was carefully read and edited accordingly.

Reviewer 4 Report

The manuscript by Ramos et al. fit the scope of this Journal. It is an overview of G-quadruplex(G4) ligands based on porphyrins. They highlighted the potential derived from the metal introduction/coordination in the selectivity of G4-target targeting over duplex structures. In addition, to better compare the metals behaviour in the G4s’ targeting, the authors decided to performe comparative experiments (determination of the binding constants by Uv-vis titration and G4-FID assay) using the same conditions to all the metal-porphyrin complexes, also adding a new Ag(II) complex.

The work includes a lot of interesting data, but I suggest reshaping the text so that it becomes more fluid.

In addition, please, pay attention to uniformate the name of the chemical compounds between the manuscript, the captions and the Figures (e.g. Figures 9, 12).

I believe that these small changes may make the paper suitable for publication in Biomolecules, MDP.

Author Response

Comment of the reviewer

The work includes a lot of interesting data, but I suggest reshaping the text so that it becomes more fluid.

Our answer

The authors thank the reviewer for the suggestion and when possible we try to reshape the comments in order to become the description more fluid. We would like to say that the article was organised chronologically and that each reported article contains several TMPyP metal complexes, that includes (or not), for instance, metals with axial ligands, and presenting (or not) conclusions related with stoichiometries and other comparative data. So, it is very difficult to separate the topics without referring the same work several times. However, with the additional information added to table 1 it is expected that the impact of metal type, ligand, interaction mode etc. on the actuation of TMPyP and analogues as GQ ligand can now be more clearly visualized. 

Comment of the reviewer

In addition, please, pay attention to uniformate the name of the chemical compounds between the manuscript, the captions and the Figures (e.g. Figures 9, 12).

Our answer

The authors thank for the advertisement and for the careful and accurate analysis of the paper. The article captions were changed and uniformized accordingly.

Round 2

Reviewer 3 Report

The manuscript has been improved significantly. The authors have addressed all the key points in the revised version. One more minor point:  please check temperature units of temperature in Table 1. Is T1/2 actually Celcius temperature? Looks like Kelvin to me.